# Insights into the coexistence of birds and humans in cropland through meta-analyses of bird exclosure studies, crop loss mitigation experiments, and social surveys

Cheng Huang[1,2,3]*, Kaiwen Zhou[1,2], Yuanjun Huang[4], Pengfei Fan[1,2], Yang Liu[1,5]*, Tien Ming Lee[1,2,5]*

1 State Key Laboratory of Biological Control, Sun Yat-sen University, Guangzhou, China, 2 School of Life Sciences, Sun Yat-sen University, Guangzhou, China, 3 State Key Laboratory of Genetic Resources and Evolution, Kunming Institute of Zoology, Chinese Academy of Science, Kunming, China, 4 School of Life Sciences, Guangzhou University, Guangzhou, China, 5 School of Ecology, Sun Yat-sen University, Shenzhen, China

* huangchengeco@163.com (CH); liuy353@mail.sysu.edu.cn (YL); leetm@mail.sysu.edu.cn (TML)

**Data Availability Statement:** All relevant data are within the paper and its Supporting Information files.

## Abstract

Birds share lands with humans at a substantial scale and affect crops. Yet, at a global scale, systematic evaluations of human–bird coexistence in croplands are scarce. Here, we compiled and used meta-analysis approaches to synthesize multiple global datasets of ecological and social dimensions to understand this complex coexistence system. Our result shows that birds usually increase woody, but not herbaceous, crop production, implying that crop loss mitigation efforts are critical for a better coexistence. We reveal that many nonlethal technical measures are more effective in reducing crop loss, e.g., using scaring devices and changing sow practices, than other available methods. Besides, we find that stakeholders from low-income countries are more likely to perceive the crop losses caused by birds and are less positive toward birds than those from high-income ones. Based on our evidence, we identified potential regional clusters, particularly in tropical areas, for implementing win-win coexistence strategies. Overall, we provide an evidence-based knowledge flow and solutions for stakeholders to integrate the conservation and management of birds in croplands.

## Introduction

Agriculture development is vital for reducing poverty and hunger [1], yet many agricultural practices substantially threaten over a quarter (27%) of all assessed 134,000 species (https://www.iucnredlist.org/). To minimize such threats, governments around the world have legally protected over 15% of the Earth's land from human-oriented development for wildlife (land-sparing scenario) [2–4]. Meanwhile, humans and wildlife often share lands in or at the edge of agricultural landscapes (land-sharing scenario), where both ecosystem services and disservices of wildlife occur to humans [5,6]. Ecosystems services are the benefits that people derive from

**Funding:** This study was funded by the State Key Laboratory of Genetic Resources and Evolution (GREKF20-03 to CH), National High-level Talent Program of China (41180953 to TML), and the DFGP Project of Fauna of Guangdong-202115 from Science and Technology Planning Projects of Guangdong Province (2021B121210002 to YL). The funders had no role in study design, data collection and analysis, decision to publish, or preparation of the manuscript.

**Competing interests:** The authors have declared that no competing interests exist.

ecosystems, such as food provisions, crop pest controls, and recreations [7–10]. In contrast, ecosystem disservices are harmful to humans, such as livestock depredation and crop damage to farmers [11–14]. In the land-sharing scenario, wildlife-friendly agriculture is widely encouraged for a better coexistence and includes two major strategies in production: (i) integrating the beneficial biological service and (ii) mitigating the harmful impacts from the disservice [15,16].

Birds, distributed across the globe, share lands with humans at a substantial scale and provide diverse provisions (e.g., meat and eggs) and cultural services (e.g., birdwatching and ecotourism) to humans [7,9]. Also, birds affect crops through direct (e.g., crop consumption) and indirect (e.g., pest control) processes [8,10,17]. Thus, birds were often treated as a key indicator taxon of agricultural sustainability in global biodiversity conservation framework (e.g., Convention on Biological Diversity). Most birds prey on invertebrates, including not only herbivore arthropods (e.g., moth larvae and stink bugs) that consume plants but also predatory arthropods (e.g., spiders and ants) that predate these pest herbivores [18,19]. Birds have been shown to suppress the herbivore arthropods in wild plants and several crops [17,20]. Furthermore, birds not only provide pollination services for many plants [21], they also directly caused losses to farmers by consuming crops including cereals and fruits [22–25]. To our knowledge, there is presently a limited number of studies synthesizing evidence to show that the indirect services actually cascade down to crop production (but see [20,26]).

In reality, recognizing a service or value from wildlife by humans is often a prerequisite for tolerating a disservice [27–29]. In a complex bird-arthropod-crop system, the indirect service from birds (e.g., pest control) may be "socially" concealed by the direct crop damage since farmers mainly care about what they can observe directly [30,31]. As a result, birds, which provide potential beneficial services to crops, are often either improperly or mistakenly eliminated by poisoning, shooting, and entangle netting, for example, approximately 20% of bird individuals caught in rice fields were nontarget bycatch in the central plains of Thailand [32–34]. Therefore, systematic evaluations of the perception and attitude of crop producers toward birds and the effectiveness of nonlethal mitigating measures (e.g., using scaring devices and changing farming practices) are essential for bird-friendly policy-making and local practices.

Here, we provide a synthesis of the extent of human–bird coexistence in croplands from the direct (seen) damage to ecological cascades to crops, the human perceptions on the biological processes, and possible solutions to address the observed disservice by birds (Fig 1a). Using standard review protocols [35], we collected and screened relevant literature ("Search rules" and Fig A in S1 Text) and compiled 4 complementary global datasets (S1–S4 Data). Then, we used standard meta-analysis approaches [36], when possible, to understand four fundamental aspects of the complex system: (i) what is the spatial extent of the potential direct (seen) disservice or crop damage by crop-consuming birds (Fig 1b); (ii) whether birds can provide a net benefit to crop productions by accounting the direct (e.g., consume crops) and indirect effects (mainly referring to pest control) of birds (Fig 1c); (iii) if disservice exists, which nonlethal mitigation measures are effective (Fig 1d); and (iv) what are the public general perception and attitude towards birds (Fig 1e). Based on our results, we proposed a strategy for a better coexistence and discussed where to take possible actions at a regional scale (Fig 1f). Our study provides scientific evidence for multiple stakeholders to better integrate key conservation strategies and bird management in croplands.

## A tenth of all bird species are known to consume crops

Birds affect crops directly (e.g., consume crops) that can be deemed as a disservice by local farmers. To understand the spatial extent of the potential direct (seen) disservices, we identify crop-consuming species by evaluating all the diet-related descriptions in the Birds of the

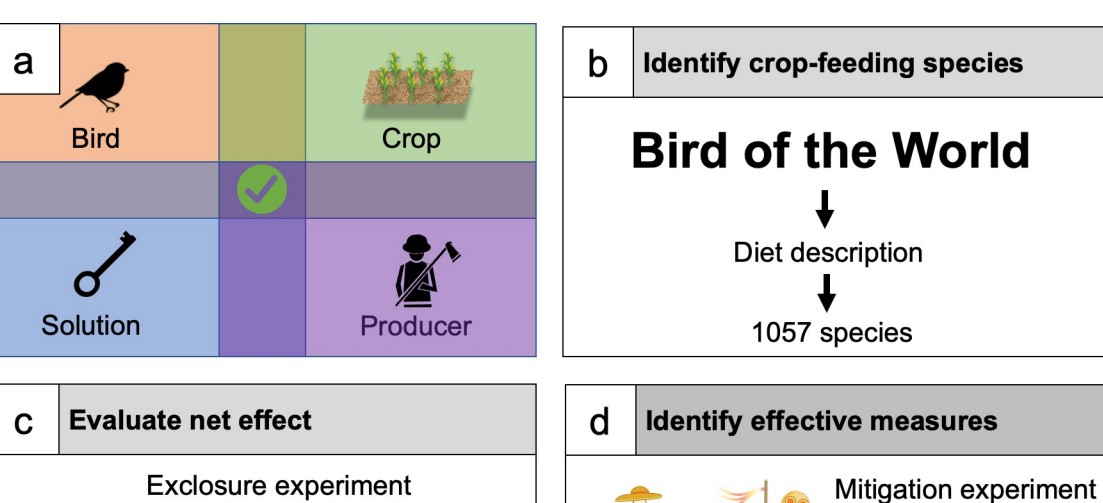

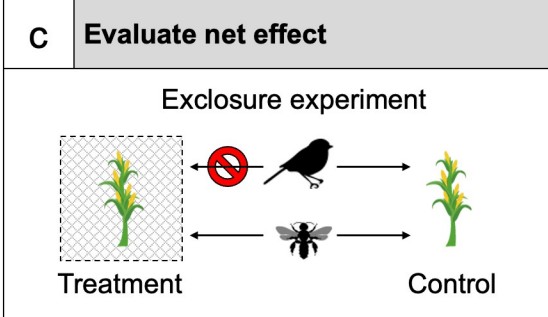

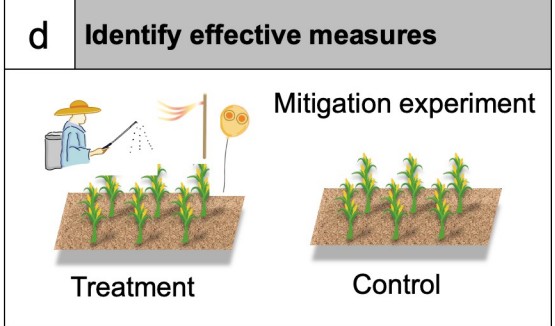

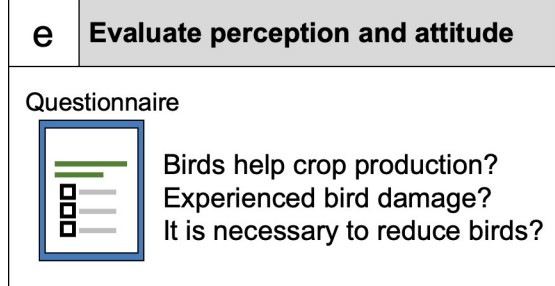

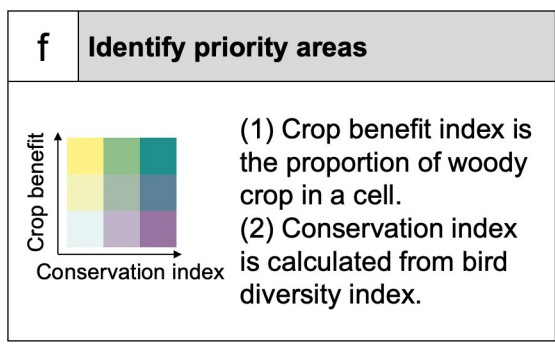

**Fig 1. A methodological overview for synthesizing evidence of human–bird coexistence in croplands.** (a) Key components of human–bird coexistence system in croplands. (b) To identify crop-consuming species, all diet-related descriptions in the Birds of the World were compiled. (c) To evaluate the net effect of birds on different crop productions, we analyzed data from exclosure experiments, of which the access of birds to crops was intentionally manipulated. (d) To offer potential solutions for reducing crop loss from birds, we analyzed data from mitigation experiments to evaluate the effectiveness of available measures. (e) To explore human perceptions and attitudes toward birds, we evaluated data from different social surveys. Based on the evidence from (b), (c), and (d), we attempted to identify the priority areas to take actions for encouraging the coexistence. Such areas are characterized by a higher proportion of crop that benefits from birds' service and higher bird conservation value (f).

World (BoW, hereafter), which is a comprehensive and widely used document about bird life history at a global scale [37–40]. For each species, the BoW provided a specific "Diet and Foraging" section, including the specific crops consumed by birds.

The BoW reported that approximately 10% of all bird species (1,057 of >10,000 species; Table A in S1 Text) consume crops (Table B in S1 Text). The major groups of birds reported to consume woody (564 species) and herbaceous (708 species) crops are generally similar (Fig B in S1 Text), including the parrots (Psittacidae), finches (Fringillidae), pigeon or dove (Columbidae), ducks or geese (Anatidae), and New World blackbirds (Icteridae). About 11% of these crop-consuming species are threatened (i.e., Critically Endangered, Endangered, and Vulnerable; Fig 2a), which is similar to the overall threatened status of birds (i.e., 12.9%). In

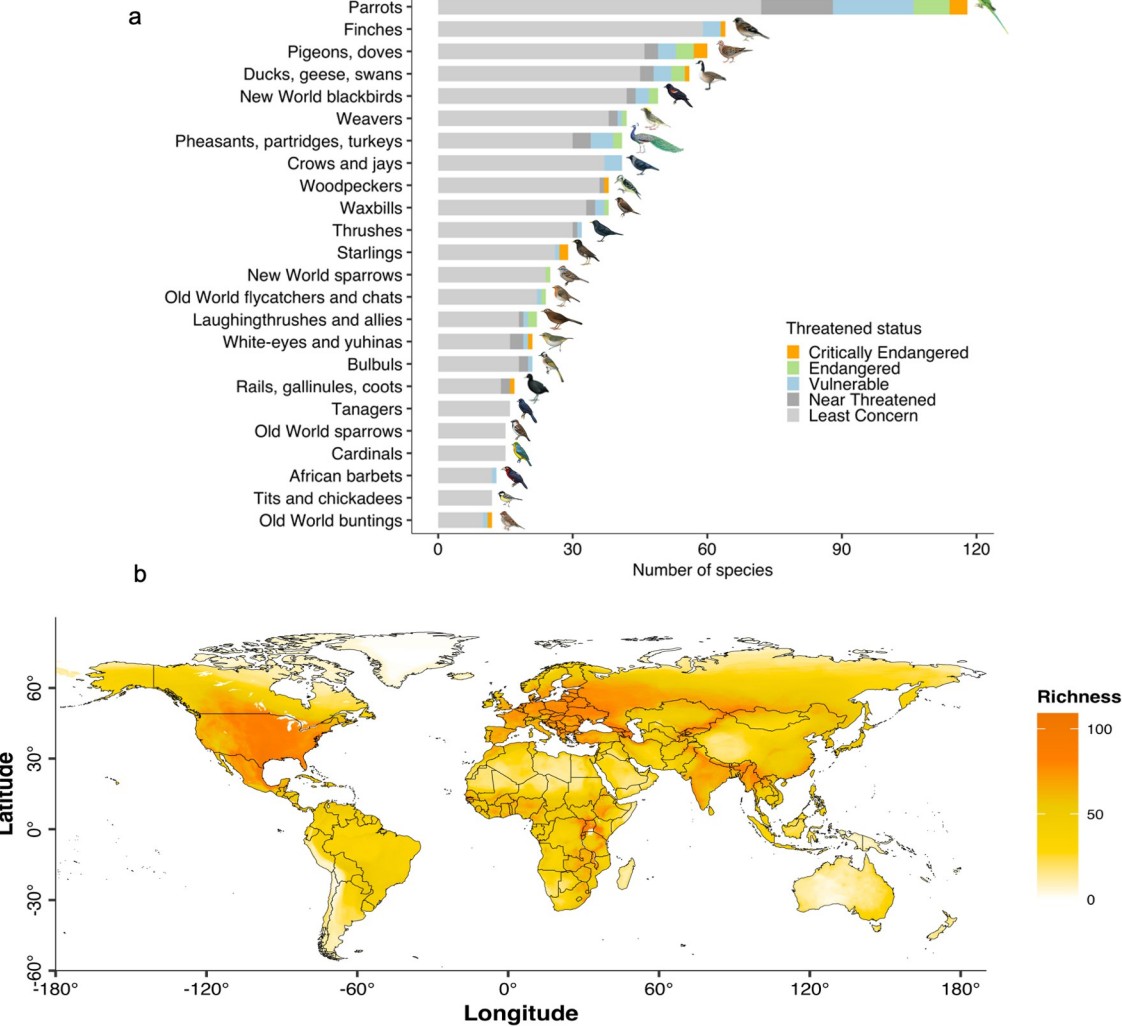

**Fig 2. Number of bird species reported to consume crops and the overlay of their range distributions.** The top barplot (a) shows the number of bird species reported to consume crops of taxonomic families by the IUCN threatened status (the data underlying this figure can be found in S1 Data). Only families with the top 25% number of species were plotted. The bottom map (b) shows the overlay of the range distribution maps of 1,057 crop-consuming bird species. The base map of country boundaries was from https://www.naturalearthdata.com/.

general, the crop-consuming birds are widely distributed, with higher richness reported in Europe, North America, East Africa, India, and Southeast Asia than in other regions (Fig 2b).

## Birds often benefit woody, but not herbaceous, crops

To estimate the net outcome of direct (e.g., consume crops) and indirect effects (e.g., pest control) of birds on crop production, we compiled a global dataset on 19 crops consisting of 158 exclosure experimental comparisons from 40 studies (Fig C in S1 Text). In these experiments, the access of birds to crops was intentionally manipulated using grids of proper mesh size, which allow the access of arthropods but exclude the birds [41]. Since there is no universal measurement for crop production, we calculated a unit-less standardized effect size (i.e., Hedges' $g$) for each experimental comparison (158 effect sizes in total). For an intuitive interpretation, we treated the open crop access to birds as experimental treatments and the

exclosures as controls; i.e., a positive effect size indicates that birds increased crop production. To estimate the overall effect, we calculated an averaged effect size across experimental comparisons, namely, a weighted mean of the individual effect sizes, using multilevel (random effect) models, in which each effect size is weighted by the inverse of within-study plus between-study variance [42,43]. Moreover, we used multiple methods to check publication bias, which occurs when the included case studies are selectively published based on their results including the significance of *p*-values, the magnitude of effect estimates, and sample sizes.

To estimate the overall effects of birds on crop production, we constructed a multilevel model to estimate an overall effect size using the 158 effect sizes. The overall effect size was marginally small (0.06, 95% confidence interval: −0.19 to 0.32, *p* = 0.62), and the Q-test [44] suggests a significant heterogeneity among the effect sizes of experimental comparisons (*p* = 0.01) (Table C in S1 Text).

To understand the source of heterogeneity among the 158 effect sizes, we identified key variables influencing the direction of the effect size (i.e., positive or negative) using a conditional inference tree, which performs binary splits by statistically significant variables [45]. Among the variables of stem types (i.e., woody and herbaceous crop), food types (i.e., coffee and cacao, fruit, treenut, cereal, vegetable, and oilcrop), climatic regions (i.e., tropical and nontropical region), insecticide use (i.e., yes or no), herbicide use, and focal animals (i.e., bird in general, herbivorous wildfowls, and bird and bat), the inference tree revealed that the stem type is the only significant variable in predicting the direction of the effect size (*p* < 0.001) (Fig D in S1 Text). For woody crops, the proportion of positive effect sizes (65%) is much higher than that of the herbaceous crops (18%) (Fig D in S1 Text). Meanwhile, the Q-test of the 158 effect sizes indicated that crop stem is a key variable affecting the heterogeneity of the results (*p* < 0.001). As such, we analyzed the data of woody (101 effect sizes) and herbaceous (57 effect sizes) crops separately (Figs 3 and 4).

To estimate the overall effect of birds on woody crop production, we constructed a multilevel model to estimate an overall effect size of the 101 effect sizes (Fig 3a), in which the term "bird" mainly refers to Passeriformes or perching birds. The overall effect size is 0.46 (CI: 0.23 to 0.70), and this result is not affected by publication bias (regression test: *p* = 0.24; Kendall test: *p* = 0.67) (Fig E and Table C in S1 Text). Then, we set potential moderators as fixed-effect factors in the model of 101 effect sizes and used Q-test to check the significance. We found that the overall effects from birds on woody crops were not significantly heterogeneous by climatic regions ($Q_m$ = 1.81, *p* = 0.18), food types ($Q_m$ = 0.61, *p* = 0.89), focal animal groups ($Q_m$ = 1.12, *p* = 0.29), use of insecticide ($Q_m$ = 0.06, *p* = 0.81) and herbicide ($Q_m$ = 0.19, *p* = 0.67), regional-level bird richness ($Q_m$ = 0.03, *p* = 0.86) and crop types ($Q_m$ = 4.59, *p* = 0.71) (Fig 3b–3d and Table D in S1 Text). In our dataset, birds had overall positive effect sizes on 6 out of 8 woody crops, particularly for coffee (0.70, 0.24 to 1.16) and apples (0.66, 0.06 to 1.27) (Table D in S1 Text). Furthermore, we counted the number of studies that illustrated the birds' effects on crop production that might be due to certain landscape variables. The counting summary showed that the effects of landscape variables (e.g., distance to edge and distance to or coverage of primary forest or habitat patch) on the birds' impact were insignificant for over 67% of the variables (Table E in S1 Text). In sum, birds have a generally positive net effect on woody crops, but negative cases also existed.

To estimate the overall effect of birds on herbaceous crop production, we constructed a multilevel model on 57 effect sizes. These experiments often focused on the taxonomic families of conflict-prone birds (Fig 2a), including perching birds (e.g., New World blackbirds, starlings, pigeons, parrots, and crows) and herbivorous wildfowls (e.g., cranes, ducks, and geese). The overall effect size is −0.48 (CI: −0.80 to 0.16), which appeared to be affected by publication

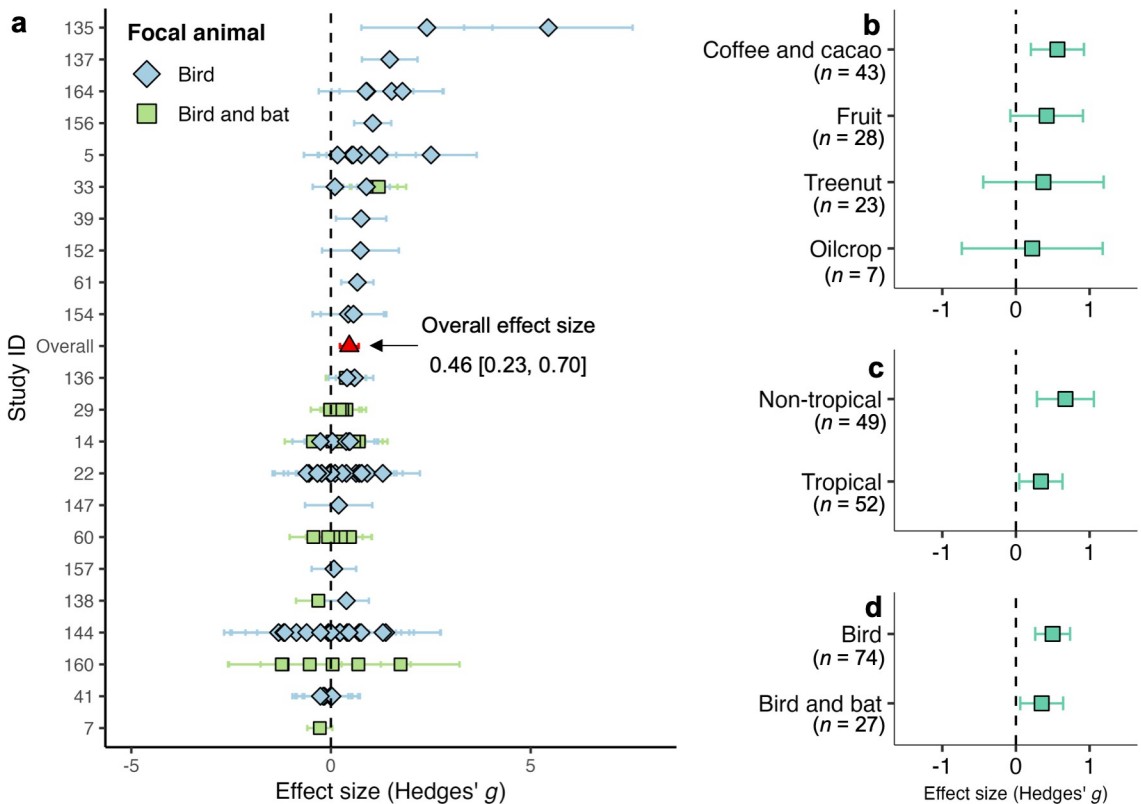

**Fig 3. Effect of birds on woody crop production.** (a) The overall effect size (in the red triangle) and the 101 effect sizes from case studies (b) by food types; (c) by climatic regions; (d) by focal animals. The term "bird" mainly refers to Passeriformes or perching birds. In all exclosure experiments, birds were the focal animal of interest, but some experiments necessarily excluded leaf-gleaning bats (d). Plotted are the effect sizes and corresponding 95% confidence interval. A positive value of effect size means that the presence of birds increases crop production and vice versa. The value *n* is the number of effect sizes per variable of interest. The data underlying this figure can be found in S2 Data.

bias (regression test: $p < 0.0001$; Kendall test: $p = 0.001$) (Fig 4a and Tables C and E in S1 Text). Thus, we performed a "trim and fill" as a sensitivity analysis, which removes funnel asymmetry by filling or mirroring the negative side of funnel plot [46]. The adjusted overall effect size is now −0.22 (CI: −0.45 to 0.01) (Table C in S1 Text).

Further, the overall effect size was influenced by food types ($Q_m = 10.14$, $p = 0.02$), climatic regions ($Q_m = 6.76$, $p = 0.01$), and crop types ($Q_m = 36.94$, $p = 0.00$). Our results showed that these birds led to crop production losses for cereals and oilcrops, particularly so for sunflowers (−1.79, −2.89 to −0.68), soybean (−0.86, −1.67 to −0.04), rice (−0.85, −1.36 to −0.16), and wheat (−0.6, −1.04 to −0.16) (Fig 4b and Table D in S1 Text). For climatic subgroups, birds generally have a significant negative effect on the herbaceous crops in nontropics (−0.62, −0.93 to −0.32), but this is not observed in the tropics (0.40, −0.31 to 1.10) (Fig 4c). Besides, we found that the overall effects of birds on herbaceous crops were not significantly varied by focal animals ($Q_m = 4.47$, $p = 0.11$), the use of insecticide ($Q_m = 0.14$, $p = 0.71$), and herbicide ($Q_m = 1.76$, $p = 0.18$) (Fig 4d and Table D in S1 Text).

## Nonlethal mitigating measures are effective to reduce crop losses

Based on our results from Figs 3a and 4a, birds could lead to crop losses in some cases, particularly for the herbaceous ones; thus, mitigation measures to reduce crop losses are essentially

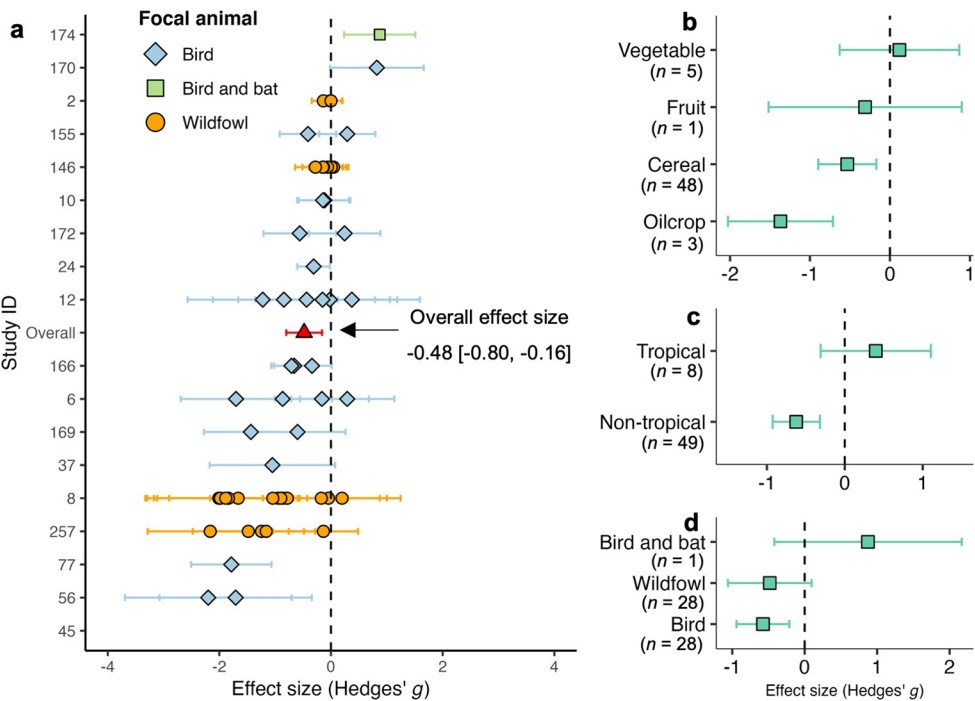

**Fig 4. Effect of birds on herbaceous crop production of. (a)** The overall effect size (in the red triangle) and the 57 effect sizes from case studies **(b)** by food types; **(c)** by climatic regions; **(d)** by focal animals. The term "bird" mainly refers to Passeriformes or perching birds. In all exclosure experiments, birds were the focal animal of interest, but some experiments necessarily excluded leaf-gleaning bats **(d)**. Plotted are the effect sizes and corresponding 95% confidence interval. A positive value of effect size means that the presence of birds increases crop production and vice versa. The value *n* is the number of effect sizes per variable of interest. The data underlying this figure can be found in S2 Data.

needed for a better human–bird coexistence. To evaluate the current set of measures mitigating the crop losses, we compiled the most comprehensive dataset of 114 experimental comparisons (or 114 effect sizes) from the agricultural fields of 18 crops from 48 studies for another meta-analysis (Fig C in S1 Text), a method similar to that of the above exclosure experiments. Here, we considered mitigation measures as our experimental treatments. We then calculated the effect size (i.e., Hedges' *g*) between the treatments and controls. When the effect size is positive, there is evidence that mitigation measures increased the crop productions or reduced the crop losses.

All of these experiments are conducted for evaluating short-term impact of technical measures. According to the crop planting stages, we classified the measures as either for the sow-stages (just for the sowing period) or all-stages (for all periods) (Table 1). A measure often affects animals through behavior stimuli (e.g., odor, taste, sound, and visual) but often is multi-faceted. Therefore, we classified the measures into eight broad categories based on management practices rather than the stimuli, which included tape/ribbon/flag, scaring model, physical barrier, sound, repellent, bird perch, herbicide to roost, and sowing practice (Table 1).

Overall, mitigating measures reduced the crop losses from birds (effect size 1.20, 95% confidence interval: 0.81 to 1.60). We also constructed a multilevel model using data of 93 effect sizes of all-stages measures. The overall effect size is 1.06 (0.67 to 1.46), which is robust to publication bias and did not significantly vary among measures (*p* = 0.48). Among all-stage measures, the use of tape, ribbon or flag (1.76, 0.97 to 2.54), scaring model (1.23, 0.35 to 2.12), net (1.04, −0.17 to 2.25), sound (0.98, −0.01 to 1.98), and repellent (0.89, 0.36 to 1.42) (Fig 5a and Table F in S1 Text) are generally effective (in decreasing order of impact). However, there is

**Table 1. Mitigating measures to reduce the crop losses caused by birds.**

| Stage | Category | Measure | Description |
|---|---|---|---|
| All | Tape/ribbon/flag | Tape/ribbon/flag | Mounting tape, ribbon, or flag to scare birds |
| | Scaring model | Balloon/kite | Balloon and kite mimicking predators through shape and eyeball |
| | | Predator model | Artificial predator model mounted on pole |
| | Physical barrier | Net | Using nets to cover the tree canopy or fruit |
| | Sound | Bio-sound | Playing alarm, distress, and fearsome sound of animals |
| | | Loud sound | Playing louds sound to scare birds |
| | Repellent | 4-aminopyridine bait | $C_5H_4N–NH_2$<br>Baited seed causes convulsions of birds |
| | | Methiocarb | $C_{11}H_{15}NO_2S.$<br>Seeds or fruits treated with methiocarb |
| | | Methyl anthranilate | $C_8H_9NO_2.$<br>Seeds or fruits treated with methyl anthranilate |
| | Bird perch | Bird perch | Providing a standing place for insectivore or aggressive birds |
| | Herbicide to roost | Habitat alteration | Glyphosate herbicide aerially applied to blackbird habitat |
| Sow | Sow practice | Sow deeper | Sowing seeds deeper than the usual |
| | | Sow late | Sow late according to other resource availability and its timing perceived by animals |
| | | Denser seed | Dense sowing of cheap seeds |
| | Physical barrier | Straw barrier | Covering sowing seeds with crop straw |
| | Repellent | Anthraquinone | $C_{14}H_8O_2.$<br>Seed or seedling treated with anthraquinone causes post-ingestional distress of birds |

no significant difference in the effectiveness among these five broad management categories ($p = 0.51$). With a relatively large sample size available in the repellent subcategory, we constructed a multilevel model using data of 32 effect sizes of the all-stages repellents. The overall effect size was 0.85 (CI: 0.39 to 1.32) and was significantly affected by repellent types ($p < 0.005$). The results suggest that methiocarb (1.23, 0.72 to 1.74) and 4-aminopyridine bait (0.74, 0.01 to 1.47) are effective (Fig 5b), while methyl anthranilate, a widely available repellent in the market, is generally ineffective. Among sow-stage measures, sow practice (7.94, 4.98 to 10.90) and straw cover (7.76, 1.87 to 13.64) are far more effective than anthraquinone (1.76, −0.70 to 4.23) (Fig 5c). Field applications of effective measures are illustrated in Fig 5d.

## Social complexities in human–bird coexistence in croplands

To understand the public perception and attitude on birds, we compiled a dataset including 39 independent surveys for over 25,000 respondents from 24 countries (Fig C in S1 Text). These surveys were mostly based on questionnaires retrieved from study areas. We extracted data on the proportion of respondents who perceived the birds' service or disservice clearly directed at crops (e.g., "birds benefit crops?" and "birds cause damages to crops?"). For the attitude, we treated the proportion of respondents who like or favor a species to "increase" or remain "stable" as the positive and vice versa [47]. We compiled four survey-level variables about the economic status of the participating country (i.e., high- or low-income) [48], focal bird groups (Fig 6), focal stakeholders (i.e., urban residents, rural residents, and the mix of these two groups), and climatic regions (i.e., tropics and nontropics).

While this may be a simplistic outcome influenced by multifaceted social factors, these surveys revealed that the average proportion of respondents having a positive attitude toward birds (67% ± 28%) is at least twice as much as those with a negative attitude (29% ± 28%). Although pest control service offered by birds is widely recognized in global biodiversity conservation agreements, only 61% (SD = 28%) of the rural residents from eight surveys thought

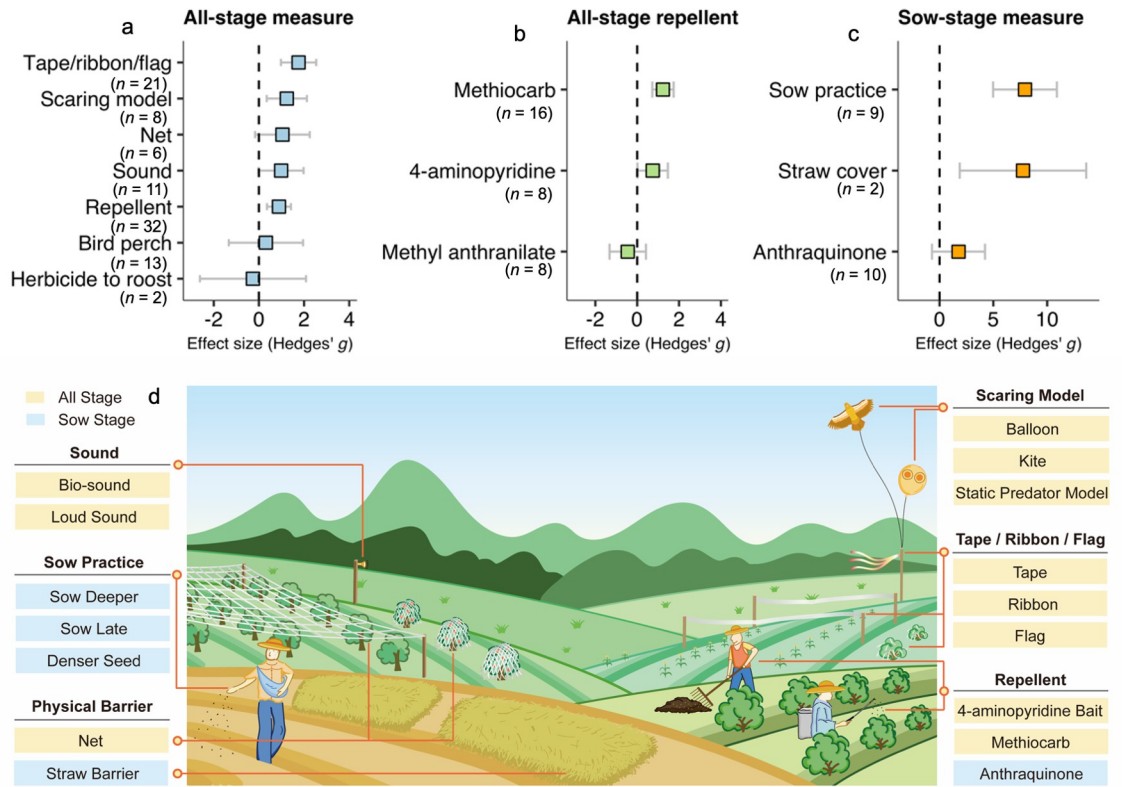

**Fig 5. Effectiveness of mitigating measures in reducing crop loss from birds.** (a) By the category of measure applied to all stages; (b) by repellent applied at all stages; (c) the category of measure applied only at sow stage. Examples of field applications are illustrated (d). Plotted are the overall mean value and 95% confidence interval of the effect size. A positive value of effect size means that a measure reduced the crop loss and vice versa. The data underlying this figure can be found in S3 Data.

that birds could benefit crops and 49% (SD = 30%) of respondents from 22 surveys perceived the disservice to crops.

To explore the influence of the four survey-level variables, we averaged the generalized linear mixed-effects models with comparable supports (ΔAICc < 2) for the perceived disservice from birds and positive attitude towards them (Table G in S1 Text). These two averaged models revealed that respondents from low-income countries are more likely to perceive the disservice ($\beta = 3.00$, $p < 0.01$; Fig 6a) and are less positive toward birds ($\beta = -1.98$, $p = 0.03$; Fig 6b) than those from high-income ones (Tables H and I in S1 Text). Meanwhile, the perceived disservice and positive attitude often varied among focal bird groups (Fig 6).

## Implications for future action and research

### A win-win strategy for promoting coexistence

Based on our meta-analyses and other scientific evidence from the socioecological perspective: (i) birds can benefit woody crops but often cause damage to herbaceous crops; (ii) many non-lethal short-term mitigating measures are effective in reducing the disservice but do not eliminate it; (iii) only 61% of those interviewed think that birds could benefit crops; and (iv) programs linked to short-term economic benefits appear to have a higher adoption rate of sustainable practices [49–51]. We propose a win-win strategy for a better human–bird coexistence in croplands. This requires that the relevant conservation and agriculture sectors to actively

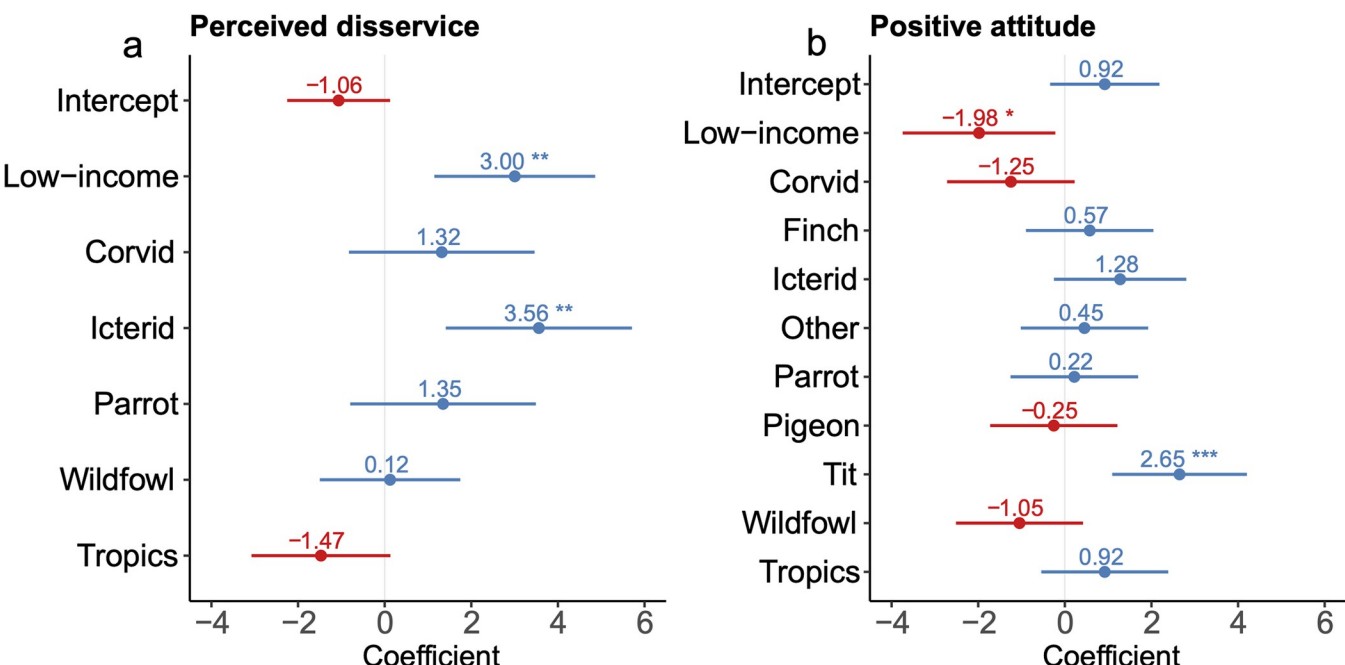

**Fig 6. Coefficients estimated by the averaged generalized linear mixed-effects models of comparable supports for the perception and attitude toward birds.** Respondents from low-income countries are more likely to perceive the disservice from birds (a) and are less positive towards birds (b) than those from the high-income ones. For the comparison among focal bird groups, the base intercept is "bird in general," which refers to the surveys that only specified the level to "birds"; and the "other" birds include woodpeckers, robins, and hummingbirds. The asterisk indicates the estimation is statistically significant (95%). A blue dot means the coefficient is larger than zero, and a red dot means the coefficient is smaller than zero.

communicate the evidence of the benefits to woody crops, so as to promote the utility of birds when training producers to reduce crop loss using more bird-friendly (or nonlethal) measures.

## Prioritize such efforts in tropical regions

Through a global strategic mapping and overlay exercise, we attempted to prioritize areas, in spite at a coarse level, to take actions at a regional scale. Such areas are characterized by more crop benefit from birds and higher bird conservation values. We treated a higher proportion of woody crops as more crop benefit from birds, since our results support that birds often benefit woody, but not herbaceous, crops. First, we overlaid the global production maps (obtained from Earthstat in 5 minutes by 5 minutes resolution) of all and woody crops, respectively (Fig F in S1 Text), and then calculated the proportion of woody crops in a cell [52]. Next, from a myriad of indices of bird conservation value [53], we selected the two most straightforward and practical indices for policymakers, namely, the richness of all threatened bird species and the richness of all bird species (Fig G in S1 Text). We considered all bird species (not just crop feeders) largely because when distribution ranges overlap the birds may be directly or indirectly impacted by the crop production activities. We then developed an integrative conservation value index by multiplying the standardized value (range from 0 to 1) of the selected two indices (Fig G in S1 Text). We assumed that an area with higher richness of bird species and higher richness of threatened bird species should be with a larger conservation value. Finally, we created the bivariate maps of crop benefit index and the indices of bird conservation value in tercile increments (33%) and treated the top 33% overlaps as the priority areas [12,54].

Because the prioritized areas are similar across three indices of bird conservation values (i.e., richness of threatened species, richness of all birds, and the integrative index based on the

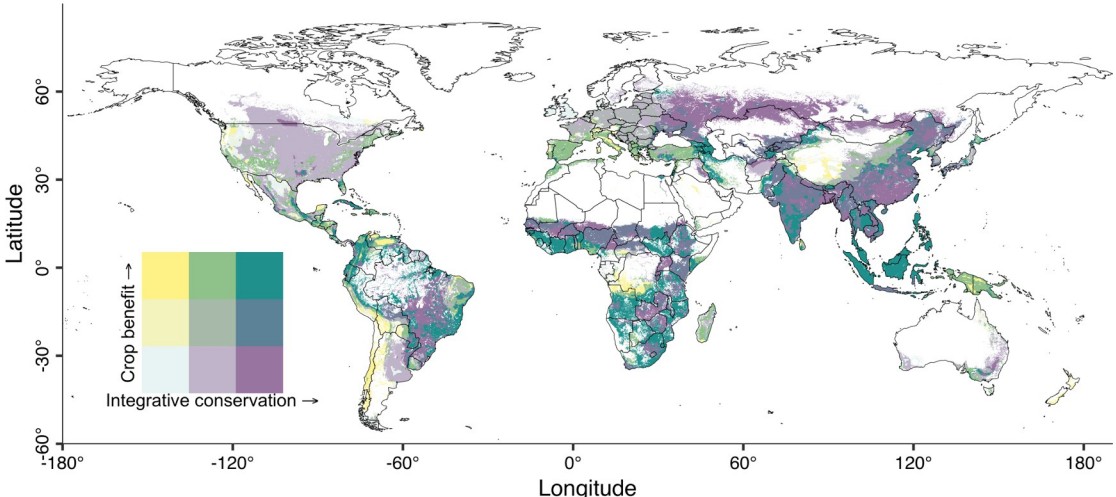

**Fig 7. The prioritized areas for efforts to make a better human–bird coexistence in croplands.** The bivariate maps are the overlay of the index of integrative conservation index of birds and crop benefit from birds in tercile increments (33%). The greener color of the bivariate maps (top-right of the legend) represents a higher integrative conservation value and more woody crops that may benefit from birds. The base map of country boundaries was from https://www.naturalearthdata.com/.

previous two) (Fig H in S1 Text), we only reported the results from the integrative conservation value (Fig 7). The priority areas (shades of green at the top right corner of the legend) are fragmentedly located across the globe but are clustered in most tropical regions, including India, Southeast Asia, sub-Saharan Africa and parts of Amazonia, and Southern Africa (Fig 7). Drawing on other insights, in these regions, it becomes critical to communicate our evidence to crop producers, such as the benefits to woody crops, as well as bird-friendly mitigating measures to reduce crop loss, which may benefit both crop production and bird conservation.

## Study limitations

Humans are sharing and will share more space with birds across the world. Thus, ensuring a better coexistence is a real-world goal for bird conservation in this land-sharing scenario. Based on our research and its limitations, we suggest to pay more research efforts on the coexistence in the following aspects.

First, we attempted to depict the extent of the potential conflict by identifying crop-consuming species and overlaying their species range distributions. Despite being the most comprehensive document on life histories of birds, the diet descriptions in the BoW may suffer from unsystematic biases across regions due to uneven knowledge (e.g., the South American region with one of the highest bird diversities) [55] and for many other reasons, including the lack of field observations for some species and the reporting preference of certain authors. Moreover, for the map (Fig 2b), we also assumed that a bird's diet is homogenous across its entire range though we are aware that it can vary by sites and subspecies [56–58]. Thus, as a future evaluation, integrating other regional lists of crop-consuming birds [55,59,60] and identifying the geographical dietary variation are needed to provide a more accurate map of the potential conflicts.

Second, our study shows that birds have a generally positive effect on woody crops but also lead to crop losses in some cases, particularly for herbaceous ones in nontropical regions. These complex patterns may jointly be contributed by habitat alterations (e.g., the common crane *Grus grus* selectively used arable lands for foraging around small protected areas) [61],

animal traits (e.g., a greater extent of conflict is often related to dietary generalists like red-winged blackbirds *Agelaius phoeniceus* and the tolerance of birds to human are higher for species with larger flock size and smaller body size) [55,62,63], and human behaviors (e.g., less harmful behaviors towards many species) [64]. These factors also contributed to human conflicts with large mammals, such as elephants, carnivores, and primates [5,64,65]. Thus, a deeper understanding of how multifaceted factors shape the variations of foraging strategies at a species and community level is critically needed, especially in agricultural fields with herbaceous crops in biodiversity-rich tropics.

Third, we collected relevant literature from three important topics of conservation biology: ecosystem services of wildlife, mitigation of wildlife disservice, and human attitudes toward wildlife. All three topics have at least a publication history over a few decades. Our temporal analysis showed that only studies about mitigation are not increasing (Fig 8 and S5 Data). This is even though we are in an era of literature explosion [66]. Furthermore, it is widely recognized that variations of effectiveness across sites and habituation are two major challenges for many kinds of technical mitigating measures for vertebrates [14,67–71]. However, the research efforts on mitigations are particularly spatial biased with over 70% of the studies (*n* = 35 out of 48) being conducted in a single country (i.e., the United States of America) (Fig C in S1 Text), and these experiments were mainly conducted during one crop season. These patterns may impede the implementation of effective evidence-based solutions and building of social capacity at local scales. On the other hand, we found that the animal behavioral-based measures (e.g., scaring model and bio-sound) appear to be quite promising. As such, more studies on animal senses, including visual, auditory, and odor, are needed to inspire and innovate new measures and to improve upon the traditional ones [72–74].

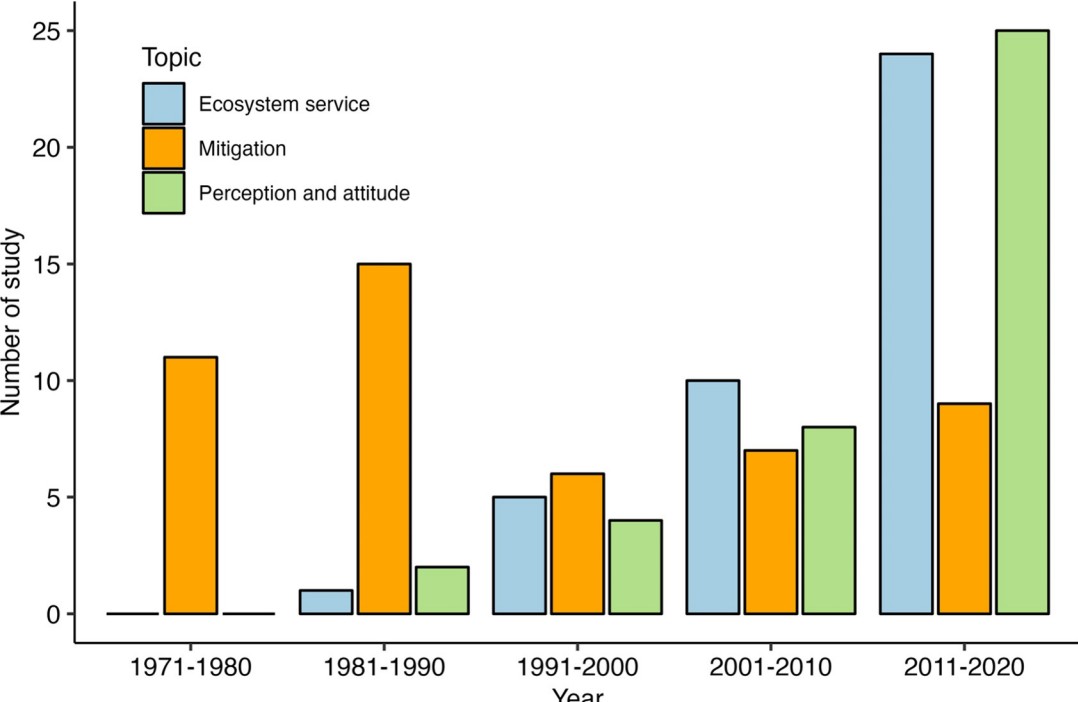

**Fig 8. Temporal trend of the number of relevant studies on ecosystem services of birds, mitigation of bird disservice, and perception and attitude towards birds.** The studies about the ecosystem service of birds (*p* = 0.02) and perception and attitude towards birds (*p* = 0.02) are generally increasing. The data underlying this figure can be found in S5 Data.

Fourth, we found that the perceived disservices from birds and positive attitude towards them are different due to the economic status of involved countries, which may be related to education, livelihood, and the extent and severity of these conflicts [47,55,75]. Nevertheless, we did not have detailed data to explore the interactions among these factors. Further, for taking actions at the local scale, studies would need to understand how multifaceted social factors (e.g., experiences, beliefs, and social norms) affect the attitudes and behaviors drawing from existing social and psychological frameworks [27,29,76,77].

## Conclusions

Our meta-analyses and synthesis uncover important insights to improve human–bird coexistence in croplands. We demonstrate that birds have a net benefit for many woody crops. Although the public is generally positive to bird conservation, the beneficial service of birds appears to be under-appreciated. Thus, improper or mistaken controls are still widespread, we uncover a number of effective nonlethal short-term measures in reducing the crop losses caused by birds. Our comprehensive study provides evidence-based information for producers, local managers, policymakers, and scientists to better integrate the conservation and management of birds in croplands to make it more bird-friendly while concurrently delivering on our food production targets.

## Materials and methods

### Data collection and compilation

To identify the species consuming 88 common crops of fruits, treenuts, vegetables, cereals, and oilcrops (Table B in S1 Text) [52], we reviewed all the diet-related descriptions of 10,721 species in the BoW (http://www.hbw.com/), which was amassed from Birds of North America, The Handbook of Birds of the World, Neotropical Birds, and Bird Families of the World and is the most comprehensive document about bird life history at a global scale [37]. The threatened status of species is based on The International Union for Conservation of Nature's (IUCN) Red List of Threatened Species 2019. The species range maps were obtained from BirdLife International (http://datazone.birdlife.org/); the production rasters of 88 crops were from Earthstat (http://www.earthstat.org/) [52].

Using standard review protocols, we performed three sets of systematic searches in Web of Science ("All Databases") to collect studies published during 1950 to 2020 in all languages for compiling the datasets of exclosure experiments, mitigating measures, and public attitudes. We defined the search rules by combinations of animal, experiment/survey method (e.g., exclosure and questionnaire), target (e.g., crop and farmer), and outcome (e.g., production and attitude), such as "bird+exclosure+coffee+yield" for searching the exclosure experiments (Tables J and K in S1 Text). All search rules were provided in "Search rules" section of S1 Text. In total, 17,959 papers were returned in the initial searches. Meanwhile, we acknowledged the potential publication bias of the Web of Science, so we also snowball-sampled studies cited by relevant papers at the screening stage to ensure the literature coverage. We included 122 scientific papers for the final meta-analyses (S6 Data); the process was recorded following the guidelines of the Preferred Reporting Items for Systematic Reviews and Meta-Analyses (Fig A in S1 Text) [35].

To be included in the exclosure or mitigation datasets, experiments had to be replicated in croplands with free-ranging wild animals. Proper data of crop production or loss (e.g., mean, its variance, and sample size) were reported for pairwise comparisons; the on-crop measurements include area-based production, weight of fruit or seed, and within-plant production, such as the yield per square meter, kg per tree, and number of shoots per square meter.

For the exclosure dataset, an experiment was typically defined by crop, site, the year it was carried out, and focal animal (i.e., bird, bird and bat, and herbivorous wildfowls). In all

experiments, birds were the focal taxon of interest, but some experiments necessarily excluded leaf-gleaning bats [20]. We compiled data on crop type, crop management, climatic region, and location. We categorized crops by stem type because the growing habitats are often distinct between woody and herbaceous crops; to give more sensible information, we also categorized crops by general use type, i.e., coffee and cacao, fruit, treenut, vegetable, cereal, and oilcrop [52]. To investigate the effect of crop management, we coded the use of insecticide and herbicide as two binary variables. To determine whether the effect varies between climatic regions, we broadly characterized the experimental sites as located in tropics and nontropics according to a widely used climatic zone map from Meteoblue [78,79]. We compiled the location information from the reported latitudes and longitudes or the approximate center of study sites. Additionally, and where available, we extracted on-farm data on bird diversity (e.g., richness and abundance) and landscape features. However, we excluded these data for further formal meta-analysis because of the small data size and inconsistent methods across the selected papers. Instead, we extracted a broad regional-level bird richness from a global richness map of birds (excluding seabirds) (https://biodiversitymapping.org/) [80,81]. Besides, we compiled information about the effect types of landscape variables (significantly negative, nonsignificant, or significantly positive) on the experimental comparisons reported from case studies. For example, (i) if the model in cases studies used the production difference between comparisons as a response variable and a landscape variable has a significant positive coefficient and (ii) if the model used the crop production as a response variable and the interaction term of treatment type (e.g., exclosure or open) and landscape variables have a significant positive coefficient, we recorded the variable under the "positive" category.

For the mitigation dataset, we only evaluated the effectiveness of a single specific measure, so we excluded integrated measures, e.g., propane cannons plus fence; we also excluded those measures that were only tested once, i.e., decoy field, seed coat, caffeine, flutolanil, and pulegone. According to the crop planting stages, we classified the measures as either for the sow-stage or all-stage (Table 1). A measure often affects animals through behavior stimuli (e.g., odor, taste, sound, and visual) but often be multifaceted. Therefore, we classified the measures into 8 broad categories based on management practices rather than the stimuli, which include tape/ribbon/flag, scaring model, physical barrier, sound, repellent, bird perch, herbicide to roost, and sowing practice (Table 1).

For the attitude dataset, each sample was represented by a question asking respondents about their perception or attitude toward a specific bird taxon or birds in general, which have some sensible impacts on crops, e.g., pest control or crop damage. The surveys about raptors and vultures were excluded because these surveys mostly focused on raptor–livestock conflict and the vultures have no direct and straightforward impacts on crops. We extracted data on the proportion of respondents who perceived the birds' service or disservice clearly directed at crops rather than something less specific like "good for nature" or "pollinating plants." We treated the proportion of respondents who like and favor a species to "increase" or remain "stable" as positive attitudes, such as "would like to see more or current-level of geese" and "willing to support efforts to conserve the species" [47]. We also compiled the data about the focal stakeholder group (i.e., rural, urban, and the mix of the former two), climatic region, and economic status of the participating country (i.e., high- or low- income) [48].

## Standard meta-analysis

A meta-analysis is a widely used approach for integrating results from independent studies to investigate the overall patterns [36]. For the effect size metric, we used the bias-corrected Hedge's $g$, the mean difference standardized using the pooled standard deviation of the

comparison [82], which was calculated using the following equations:

$$g = J \times d$$

$$V_g = J^2 \times V_d$$

where $d$ and $V_d$ are calculated using the following equations:

$$d = \frac{X_{treatment} - X_{control}}{SD_{within}}$$

$$SD_{within} = \sqrt{\frac{(n_{treatment} - 1)S^2_{treatment} - (n_{control} - 1)S^2_{control}}{n_{treatment} + n_{control} - 2}}$$

$$V_d = \frac{n_{control} + n_{treatment}}{n_{control} \times n_{treatment}} + \frac{d^2}{2 \times (n_{control} + n_{treatment})}$$

The biased-corrected factor $J$ is defined as:

$$J = 1 - \frac{3}{4(n_{treatment} - n_{control} - 2) - 1}$$

If the data for direct effect size calculation were not reported, we extracted convertible data and then calculated $g$ using the "esc" package [83,84]. We converted effect sizes to be in the same direction by multiplying $g$ with minus one [85]. For example, while agriculture yield data were expressed in terms of production gains (5 kg versus 10 kg), crop loss data were normally expressed oppositely (50% loss versus 30% loss). Positive values of $g$ indicate that the presence of birds increases crop production or mitigating measures reduce crop loss and vice versa. We then estimated the average effect size across experimental comparisons from case studies using multilevel mixed-effects models in R "metafor" package; this model allows the specification of nested groups to control for nonindependence in the datasets due to multiple effect sizes derived from the same study [42,43,86]. Each effect size is weighted by the inverse of within-study plus between-study variance [82]. Q-test was used to test the significance of heterogeneity among comparisons that were attributed to variables [44].

We accessed the potential publication bias (or "file drawer problem") graphically (funnel plots), numerically (Rosenthal's fail-safe number), and statistically (Kendall's rank correlation) [87–89]. If the fail-safe number is sufficiently high (i.e., $> 5n + 10$, where $n$ is the number of pairwise comparisons), the significant results can be considered robust. Kendall's rank correlation examines the relationship between effect size and sample size across comparisons; significant $p$-values indicate publication bias whereby comparisons with small sample sizes are only published if they show large effect sizes [90]. If the averaged effect size is significant and the funnel plot (Fig E in S1 Text) and/or ranking correlation shows potential publication bias, we would use the "trim and fill" method as a sensitivity analysis, which recalculated the estimated average effect size by trimming the smaller comparison from the positive side and filling it or mirroring the negative side of funnel plot thereby removing funnel asymmetry [46]. This technique provides an estimate of how the overall effect size would change if we were able to incorporate all missing studies.

## Other analysis

We counted the number of studies that revealed the birds' effects on crop production that were affected (significantly negative, nonsignificant, or significantly positive) by certain landscape variables.

We calculated an arithmetic mean of the proportion of respondents with a specific perception and attitude from case studies for descriptive summaries. To explore the differences among categories of the economic status of the participating country, bird groups, focal stakeholders, and climatic regions, we first constructed generalized linear mixed-effects binomial models with these four categorical variables to model the proportion data of perception and attitude. In these models, each survey was set as a random effect and the number of respondents in the survey was used as prior weights in the fitting process [91]. For example, if 80% of 100 respondents in a survey were reported to have positive attitudes, 80 respondents will be coded as having positive attitudes and the other as nonpositive [47,91]. Then, we used "dredge" function of "MuMIn" package to rank the subletting models by the value of the Akaike's information criterion corrected for small sample size (AICc) [92]. Finally, we used the best model to examine the differences among categories; where multiple models were supported ($\Delta$AICc < 2; Table G in S1 Text), we used model averaging to better capture model selection estimation error. Since the results for the negative attitude are inversely similar to that of the positive (Tables I and L in S1 Text), we only reported the results of the positive attitude.

We tested the trend of relevant studies by a nonparametric Spearman test between the observations and time using "trend.test" function in the "pastecs" package of R [93].

## Supporting information

**S1 Data. List of species consuming crop.**
(XLSX)

**S2 Data. Exclosure experiment dataset.**
(XLSX)

**S3 Data. Mitigation experiment dataset.**
(XLSX)

**S4 Data. Attitude survey dataset.**
(XLSX)

**S5 Data. Number of studies in a 10-year scale.**
(XLSX)

**S6 Data. List of case studies.**
(XLSX)

**S1 Text. Supporting information.** Supporting text for "Search rules."
Table A. Number of crop-consuming bird species by taxonomic order and threatened status.
Table B. Eighty-eight crops were included to check whether a species is a crop-feeder or not in the Birds of the World.
Table C. Using fail-safe number, Kendall's rank correlation test, regression test, and "trim and fill" models to test publication bias. The "trim and fill" and regression test does not run for multilevel models in "metafor" package of R (rma.mv), so they were based on random model without multilevel structure (rma).
Table D. The significance test of moderators and effect size of subgroups. Q-test was used to test the significance of heterogeneity among comparisons that was attributed to moderators.

"Comparisons," "studies," and "crops" represent the number of pairwise comparisons, studies, and crop type for each subgroup, respectively. Positive value of effect size means that the presence of birds increases crop production. The lower and upper bounds of 95% confidence interval were reported.

Table E. Frequency of landscape variables showing either significant positive, nonsignificant, or significant negative impact of the birds' effects on crop production.

Table F. The significance test of moderators and effect size of subgroups. Averaged effect sizes of multilevel mixed model were calculated for the measures. "Comparisons," "studies," and "crops" represent the number of pairwise comparisons, studies, and crop type for each subgroup, respectively. Positive value of effect size means that the measures reduced crop loss caused by birds. The lower and upper bounds of 95% confidence interval were reported.

Table G. Models of comparable supports ($\Delta$AICc < 2) for the perceived disservice and positive attitude.

Table H. Estimations by the averaged generalized linear mixed-effects binomial models for the disservice.

Table I. Estimations by the averaged generalized linear mixed-effects binomial models for the positive attitude.

Table J. PICO framework for the exclosure experiment dataset.

Table K. PICO framework for the mitigating experiment dataset.

Table L. Estimations by the averaged generalized linear mixed-effects binomial models for the negative attitude.

Fig A. PRISMA flow of the meta-analyses.

Fig B. Number of crop-consuming species by taxonomic family and threatened status. The barplots show the major taxon (top 25%) consuming herbaceous (A) and woody (B) crops. The data underlying this figure can be found in S1 Data.

Fig C. Number of studies (in bracket) about (A) exclosure experiments, (B) mitigation experiments, and (C) attitude surveys by countries. A country with grey background means no relevant study. The base map of country boundaries was from https://www.naturalearthdata.com/.

Fig D. Conditional inference tree of the impact of variables on the direction of bird effect (i.e., positive and negative) on crop production. The conditional inference tree algorithm recursively tests the global null hypothesis of independence between any of the input variables and the response and then select the input variable with the strongest association to the response. The stem type (i.e., woody or herbaceous) is the only significant variable in predicting the direction of the effect size among stem type, food type, climatic region, use of insecticide, use of herbicide, and focal animal.

Fig E. Funnel plots showing the relationship between effect size and standard error of comparisons. (A) For all comparisons of exclosure experiments; (B) for the comparisons of exclosure experiments on woody crops; (C) for the comparisons of exclosure experiments on herbaceous crops; and (D) for all comparisons in mitigation dataset.

Fig F. Map of crop production and benefit index. (A) Distribution of woody crop production. (B) Distribution of total crop production. (C) Benefit index, which was calculated by the proportion of woody crop production (A) of total crop production (B) in a grid cell. The base map of country boundaries was from https://www.naturalearthdata.com/.

Fig G. Map of 3 bird conservation values. The integrative conservation value index (C) was calculated by multiplying the standardized value of the richness of threatened bird species (A), and standardized value of the richness of all bird species (B). The base map of country boundaries was from https://www.naturalearthdata.com/.

Fig H. A global mapping exercise of the prioritized areas for efforts to encourage bird-friendly croplands across 3 conservation value indices. (A) The threatened bird species richness; (B)

the richness of all bird species; and (C) the integrative conservation value index. The bivariate maps are displayed in tercile increments (33%). The dark green color of the bivariate maps (top-right of the legend) represents a higher conservation value and more benefit of woody crops from birds. Areas in white are outside of the prioritized hotspots. The base map of country boundaries was from https://www.naturalearthdata.com/.
(DOCX)

## Acknowledgments

We thank Hao Ding, Nian Guan, Suni Han, Xiaodong Li, Shaochong Peng, Mengyun Qiu, Liang Su, Bai Xiao, Liang Xu, and Qiuyang Zheng for the bird illustrations. We also thank Fangyuan Hua for comments on the manuscript.

## Author Contributions

**Conceptualization:** Cheng Huang, Pengfei Fan, Yang Liu, Tien Ming Lee.

**Data curation:** Cheng Huang, Kaiwen Zhou, Yuanjun Huang.

**Formal analysis:** Cheng Huang.

**Funding acquisition:** Cheng Huang, Yang Liu, Tien Ming Lee.

**Investigation:** Cheng Huang.

**Methodology:** Cheng Huang, Kaiwen Zhou, Yuanjun Huang.

**Project administration:** Cheng Huang, Yang Liu, Tien Ming Lee.

**Supervision:** Yang Liu, Tien Ming Lee.

**Visualization:** Cheng Huang, Yuanjun Huang.

**Writing – original draft:** Cheng Huang, Yang Liu, Tien Ming Lee.

**Writing – review & editing:** Cheng Huang, Kaiwen Zhou, Yuanjun Huang, Pengfei Fan, Yang Liu, Tien Ming Lee.

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
