## [Editor Report · Decision Letter 0]

9 Jan 2023

Dear Ming, 

Happy new year! Thank you for submitting your revised manuscript entitled "Coexisting with birds in cropland: Meta-analyses of birds exclosure and crop loss mitigation experiments, and insights from social surveys" for consideration as a Meta-Research Article by PLOS Biology. Apologies for the delay incurred over the holiday period.

Your revisions have now been evaluated by the PLOS Biology editorial staff, and I'm writing to let you know that we would like to send your submission out for re-review.

However, before we can send your manuscript back to reviewers, we need you to complete your submission by providing the metadata that is required for full assessment. To this end, please login to Editorial Manager where you will find the paper in the 'Submissions Needing Revisions' folder on your homepage. Please click 'Revise Submission' from the Action Links and complete all additional questions in the submission questionnaire.

Once your full submission is complete, your paper will undergo a series of checks in preparation for re-review. After your manuscript has passed the checks it will be sent back to the reviewers. To provide the metadata for your submission, please Login to Editorial Manager (https://www.editorialmanager.com/pbiology) within two working days, i.e. by Jan 11 2023 11:59PM.

Best wishes,

Roli

Roland Roberts, PhD

Senior Editor

PLOS Biology

rroberts@plos.org

---

## [Decision Letter · Decision Letter 1]

13 Feb 2023

Dear Ming,

Thank you for your patience while we considered your revised manuscript "Coexisting with birds in cropland: Meta-analyses of birds exclosure and crop loss mitigation experiments, and insights from social surveys" for consideration as a Meta-Research Article at PLOS Biology. Your revised study has now been evaluated by the PLOS Biology editors, the Academic Editor, and three of the original reviewers.

You'll see that the reviewers continue to be positive, but still raise a number of issues that must be addressed. I discussed the issue about the challenging structure of the paper (raised by reviewer #3) with the Academic Editor; we do not necessarily have a problem with you including the methods within each respective results section, but we do ask you to try your best to maximise the clarity of your manuscript.

In light of the reviews, which you will find at the end of this email, we are pleased to offer you the opportunity to address the remaining points from the reviewers in a revision that we anticipate should not take you very long. We will then assess your revised manuscript and your response to the reviewers' comments with our Academic Editor aiming to avoid further rounds of peer-review, although might need to consult with the reviewers, depending on the nature of the revisions.

**IMPORTANT - SUBMITTING YOUR REVISION**

*Resubmission Checklist*

*Published Peer Review*

*PLOS Data Policy*

Best wishes,

Roli

Roland Roberts, PhD

Senior Editor

PLOS Biology

rroberts@plos.org

REVIEWERS' COMMENTS:

Reviewer #1:

I appreciate that the authors have conducted a rather thorough revision, but I still have some major concerns on the methodologies used. I also found some of the authors' responses to my previous suggestions insufficient. They are largely about the lack of information provided, which I think is a critical problem in term of the study's reproducibility. Please see below my specific comments.

Figs 3 and 4. In the previous review I suggested plotting mean effect sizes for woody and herbaceous crops as well as their 95% CI, and the authors' response was "We have now plotted these details". But I can't see the plot of the mean (i.e., overall) effect size in Figs 3 and 4. They are shown as numbers, but not as dots with error bars.

L222: there is a typo "0.-0.16".

L313-325, Fig 6 (on L327): I still wonder why the authors did not perform a more rigorous multi-variate analysis here. According to their explanation, the authors seem to have simply compared averages between different categories in Fig. 6, but I still think the effect of those categories should be tested in a single model. The choice of variables is also unclear. For example, why did the authors not test the effect of the tropics vs non-tropics here? In the response letter, the authors mentioned that the small sample size prevented them from doing a more complex analysis, but the attitude dataset (Supplementary Data 4) has 50 records of positive attitude, and with this sample size I don't think it's impossible to do a multi-variate analysis. For example the meta-analysis in Fig 4 also has only 57 samples but includes multiple variables.

Regarding Supplementary Data 4, it is still unclear how positive vs negative attitudes were defined. I strongly suggest providing clearer, concrete definition in each survey. This can be done by adding another column to Supplementary Data 4 for answer categories defined as positive or negative attitude in each survey. For example, for row 2, answer "No" to question "mute swans should either be controlled or eliminated" must have been defined as positive attitude. So the new column should have "No" for row 2 and "Yes" for row 3. What is unclear to me is the definition for e.g., row 40. What kind of answers to question "dislike and strong dislike" here was defined as positive (or negative)? This needs to be clearly specified in the new column. I understand some examples are now provided in the main text but this information should be provided for all surveys, as it affects the reproducibility of the analysis.

Fig 6 (on L381). This now needs to be labelled as Fig 7 I believe.

L430-431: Cite S3 Fig here to show the spatial bias.

L446-448: I don't think this comparison (67% vs 46%) is reasonable, as it does not account for effects of any covariates. In an extreme case, for example, the difference may simply be due to a difference in locations, not due to a difference in taxonomic groups.

L490: What "All Databases" in Web of Science includes can hugely differ depending on the institute's subscription, so specify actual databases used.

L496: "Search rules" are central to this kind of study, so should be provided here (i.e., in the main text), not in the Supplementary Information.

L501: Supplementary Data 6 lacks important meta-data of each paper, such as journal, volume, and page numbers.

L504-561: In the previous review I suggested using PICO framework to describe the eligibility criteria (as is now shown in Box 5.1 here: https://environmentalevidence.org/information-for-authors/5-eligibility-screening/), which I believe is essential, but the authors do not seem to follow my suggestion.

L522: Explain how each site was defined as tropics or non-tropics. If it's based on latitudes, what was the cut-off?

L617: Again I wonder why the analysis of attitudes towards birds was based on such a simplistic approach. As the authors also admit, the response variable (attitudes) can be affected by multiple factors simultaneously, and the effect of each factor should be tested in a single multi-variate analysis.

Supplementary Information

L10: Specify who did the searches.

L11: P for PICO in ecological studies is normally Population. See: https://environmentalevidence.org/information-for-authors/2-need-for-evidence-synthesis-type-and-review-team-2/

L19-20: Give the full detail of how non-English literature was searched (e.g., database used, search terms, target languages etc), just like the search detail for English. In the response letter, the authors indicated that they also searched in Chinese using CNKI. Please also provide the detail of the Chinese searches.

L28: What is "species name of bird"? Does this mean the scientific name of each target species was used and the search was repeated for all target species? Elaborate it more. Also provide a list of target species and their scientific names.

L28-36: What does Rule 1, Rule 2, etc each mean? These search terms are normally combined using Boolean operators (AND, OR, NOT) and use in a single search - see: https://environmentalevidence.org/information-for-authors/4-conducting-a-search/. I am not entirely sure exactly how these rules were applied in this study. Please provide more explanations.

Reviewer #3:

Dear authors, 

This is the second time I review this manuscript. While you have addressed my comments, I find the paper very difficult to read because you now made the paper very long. Moreover, your paper does not have the sections that most papers have: intro/methods/results/conclusions, as you have included the results with the methods for each finding, which I find very confusing (for example lines 150-155 belong in the methods, not intro). I defer to the editor to make the call on how he would like to see the paper organized, but it still reads very choppy to me. Particularly, the limitations section is now too long and cumbersome.

Nonetheless, I recognize all the effort you have done to address the concerns by all four reviewers and so I suggest a few more minor edits.

Lines 40-43- ecosystem services are different from resources and functions. I think this sentence needs to be modified as: "Ecosystems services are the benefits that people derive from ecosystems, such as food provision, pest control, recreation". They ARE NOT functions. 

Line 49- change distributing to distributed

Line 56 change "predaceous" to predacious or predatory

Line 258= typo is "therefore" you have an extra o

Overall you need to explain what you mean by "unaffected by publication bias" in lines 268 and others. I understand this now because I read your reply letter to reviewers, but without it, it is unclear. 

I think a better header for the section "More research efforts on the coexistence are needed" is "Study limitations", because this is where you list all your limitations. 

I believe that while your limitations are honest, you have now written very long sentences and paragraphs and it made this paper very cumbersome to read. You could streamline the writing and make it more precise. For example the paragraph 446-462 is too long.

Reviewer #4:

I am impressed by the extensive revisions that the authors undertook in response to the reviewer comments. It was quite a lot of work to tackle. I believe the manuscript is very comprehensive and will be of interest to a range of researchers on agricultural sustainability, especially those working on avian ecosystem service tradeoffs and conservation. 

I only have some minor comments/suggestions for clarity or emphasis. 

L34: 27% is closer to a quarter than a third, so it seems more accurate to say something like, "over a quarter (27%) of" 

L63: I think it might make sense to say "[but see 20, 26]" rather than just citing these references because I believe both of these papers are syntheses of evidence showing indirect services cascading down to production. I think the papers [20, 26] do make the argument that more should be done, so perhaps it's fine as it is. 

L112-137: I agree with using the Birds of the World since it's the most comprehensive database on single diet items. However, I think the section could be revised slightly to acknowledge the data limitations Reviewer 3 brought up. For example, for the heading, you could say something like, "A tenth of all bird species are known to consume crops," "A tenth of all bird species have been documented to consume crops," or "A tenth of all bird species have been shown to consume crops." At L122, you could say "The major groups of birds reported to consume woody…" to acknowledge that the reports are based on what's been done to date (and what made it into the species account). Even for locations with relatively more studies like the United States, there are biases towards better documentation for common species (and very little data for rarer species). Again, at L133 in the figure legend, you could say something like "Number of bird species known to consume crops…"

Finally for the section from L112-137, I see the authors' argument to not add an analysis that is similar to another study with more robust results. However, I think the result would be of strong interest to conservationists pushing to conserve threatened species on farms who get pushback from farmers due to concerns over crop damage. I think the result could be added as a brief statement, but I defer to the authors to decide if they want to add it. 

L199: Due to the journal format of having Results before Methods, I was not sure what "the counting summary" was until I got to the Methods. It might work better to put the landscape vote counting results at the end of the current paragraph and briefly introduce that you did vote counting. 

Table 1: Please specify if the bird perches are for raptors or other birds

L306: Having read the author's response to reviewers, I understand what "birds in general" is, but I think a brief definition like "birds in general (i.e., the study only specified to the level of 'bird' in surveys)" like the other categories would be helpful for future readers

Figure 6: The asterisk for urban vs. rural blends in with the bar, so the authors may want to move it down slightly. I didn't notice the asterisk at all until I read the legend and was looking for it. 

L553: Here it says raptors and vultures are excluded, but I think perhaps perches above include raptor perches, and the search terms include raptor. I think it would be good to clarify towards the start of the manuscript if raptors are included at all, and then in the literature search section clarify again when, where, and why you excluded them since you searched them. 

L613-615: I think the vote counting method makes sense, given data availability. For people less familiar with the challenges of landscape meta-analyses, it might help to give a brief justification of why you switched to vote counting for this analysis. 

Literature search:

For the literature search, I am not sure how the "species name of bird" search was done exactly. Did you repeat this search for all 10,000 species of birds or did you just search the scientific names of the most common birds? Did you repeat the search for each species or did you search all names at once like […"species name 1" OR "species name 2" OR "species name 3"…]. The methods should clarify this. I think it was a good idea to search the scientific names as well, but I'm just not clear on exactly how you did it. 

L20 of the SI: I think "…and added only one study in Spanish" would work better based on what the authors wrote in the response to reviewers 

With the search, it makes sense that the authors started searching broader literature on fisheries and poultry then scaled back, but I think there needs to be some more justification on why the scope was narrowed.

---

## [Decision Letter · Decision Letter 2]

4 May 2023

Dear Ming,

Thank you for your patience while we considered your revised manuscript "Coexisting with birds in cropland: Meta-analyses of birds exclosure and crop loss mitigation experiments, and insights from social surveys" for publication as a Meta-Research Article at PLOS Biology. This revised version of your manuscript has been evaluated by the PLOS Biology editors, the Academic Editor and one of the original reviewers. Note that we have changed the article type from "Meta-Research Article" to "Research Article" (your study is research that happens to involve meta-analysis, rather than meta-research per se).

Based on the review, we are likely to accept this manuscript for publication, provided you satisfactorily address the remaining points raised by the reviewer and the following data and other policy-related requests.

IMPORTANT - Please attend to the following:

a) Please address the remaining requests from reviewer #1.

b) Please change the Title to "Insights into the coexistence of birds and humans in cropland through meta-analyses of bird exclosure studies, crop loss mitigation experiments and social surveys" (to avoid the colon and make the meaning more explicit)

c) Please provide a blurb, according to the instructions in the submission form.

d) Please address my Data Policy requests below; specifically, we need you to supply the numerical values underlying Figs 2AB, 3ABCD, 4ABCD, 5ABC, 6AB, 7, 8, S2AB, S5ABCD, S6ABC, S7ABC, S8ABC, either as a supplementary data file or as a permanent DOI’d deposition. We note that you have already supplied 6 supplementary data files, but these look quite “raw,” and it is unclear exactly how they relate to the Figure; please clarify and/or supply the numerical values displayed.

e) Please cite the location of the data clearly in all relevant main and supplementary Figure legends, e.g. “The data underlying this Figure can be found in S1 Data” or “The data underlying this Figure can be found in https://doi.org/10.5281/zenodo.XXXXX”

We expect to receive your revised manuscript within two weeks. 

*Published Peer Review History*

*Press*

Sincerely,

Roli

Roland Roberts, PhD

Senior Editor,

rroberts@plos.org,

PLOS Biology

DATA POLICY:

Regardless of the method selected, please ensure that you provide the individual numerical values that underlie the summary data displayed in the following figure panels as they are essential for readers to assess your analysis and to reproduce it: Figs 2AB, 3ABCD, 4ABCD, 5ABC, 6AB, 7, 8, S2AB, S5ABCD, S6ABC, S7ABC, S8ABC. NOTE: the numerical data provided should include all replicates AND the way in which the plotted mean and errors were derived (it should not present only the mean/average values).

DATA NOT SHOWN?

REVIEWER'S COMMENTS:

Reviewer #1:

I found that the authors have done an excellent job for responding to most of my comments. I especially like the newly added analysis of people's attitudes, which I believe has made the paper's message much more robust.

I now have only a few minor comments on Supplementary Information.

1. P2, L14: Replace participation with population and comparison with comparator.

2. P2, L22-24: I suggested giving the full detail of how non-English literature was searched, such as databases used, search terms, target languages etc, but the authors only provided the following explanation, which I don't think is sufficient. Please provide the full details.

"In Oct 2022, using similar approaches above, we re-searched non-English literature published before December 2020 and added only one study in Spanish in the exclosure experiment dataset."

---

## [Editor Report · Decision Letter 3]

16 May 2023

Dear Ming,

Thank you for the submission of your revised Research Article "Insights into the coexistence of birds and humans in cropland through meta-analyses of bird exclosure studies, crop loss mitigation experiments and social surveys" for publication in PLOS Biology. On behalf of my colleagues and the Academic Editor, Andrew Tanentzap, I'm pleased to say that we can in principle accept your manuscript for publication, provided you address any remaining formatting and reporting issues. These will be detailed in an email you should receive within 2-3 business days from our colleagues in the journal operations team; no action is required from you until then. Please note that we will not be able to formally accept your manuscript and schedule it for publication until you have completed any requested changes.

Sincerely, 

Roli

Senior Editor

PLOS Biology

rroberts@plos.org